# Interaction between Grasses and *Epichloë* Endophytes and Its Significance to Biotic and Abiotic Stress Tolerance and the Rhizosphere

**DOI:** 10.3390/microorganisms9112186

**Published:** 2021-10-20

**Authors:** Kendall Lee, Ali Missaoui, Kishan Mahmud, Holly Presley, Marin Lonnee

**Affiliations:** 1Institute of Plant Breeding, Genetics and Genomics, University of Georgia, Athens, GA 30602, USA; kcl58759@uga.edu (K.L.); holly.wright25@uga.edu (H.P.); 2Department of Crop and Soil Science, University of Georgia, Athens, GA 30602, USA; Marin.Lonnee@uga.edu; 3Center for Applied Genetic Technologies, University of Georgia, Athens, GA 30602, USA; kishan.mahmud25@uga.edu

**Keywords:** endophyte, *Epichloë*, forage, mechanisms, stress

## Abstract

Cool-season grasses are the most common forage types in livestock operations and amenities. Several of the cool-season grasses establish mutualistic associations with an endophytic fungus of the *Epichloë* genus. The grasses and endophytic fungi have evolved over a long period of time to form host-fungus specific relationships that confer protection for the grass against various stressors in exchange for housing and nutrients to the fungus. This review provides an overview of the mechanisms by which *Epichloë* endophytes and grasses interact, including molecular pathways for secondary metabolite production. It also outlines specific mechanisms by which the endophyte helps protect the plant from various abiotic and biotic stressors. Finally, the review provides information on how *Epichloë* infection of grass and stressors affect the rhizosphere environment of the plant.

## 1. Introduction

*Poaceae* is a large taxonomic family of grasses. Most grasses used for forage originated in Europe, Africa, and western Asia, with over 600 species from this family being used worldwide for livestock forage [1]. Typically, these grasses are cross-pollinated polyploids with complicated genomes. In America, grasses occupy over 238 million ha of land used for forage production [2]. Forage grasses are essential to livestock operations and can be native or introduced, perennial or annual, and cool-season or warm-season depending on the region and farmer preference. Pasture-fed beef and dairy are gaining popularity in the United States and are already the primary mode of livestock production in places like New Zealand, South America, and Europe. The hardiness and endurance of the grass is relied on by livestock farmers. Persistence under grazing pressure, high nutritive value, drought resistance, insect resistance and high yield are important criteria for the sustainability of forage production and climate resilience of these species. Temperate grasses, also known as cool-season, are an important class of forages that sustain the supply of most of the world’s beef and dairy products [3]. Several cool-season grasses are known to establish a fungal association with *Epichloë* species of fungi that confer abiotic and biotic stress tolerances to the grass. Biotic stressors refer to living elements and include weeds, insects, and diseases that affect plant health and compromise yield. Abiotic stressors refer to non-living elements that could harm the crop, such as drought and flood. A major difference between forage grasses and other crops is that grazing livestock is the main biotic stressor. Withstanding grazing pressures is one of the most important stress tolerances forage grasses need to possess. Producers often choose not to spray chemical pesticides for insects in grazing situations due to the cost versus return of the treatment, but spraying weeds is more common [4]. Abiotic stress tolerance is necessary for forage grasses as a less rigorously managed crop. Most pastures are subjected to drought, flood, nutrient deficiency, high temperature, frost, etc., without much intervention from producers. This leads producers to search for forage grasses that are naturally tolerant to many of these stresses. In this review, we will highlight how *Epichloë* endophytes and host grasses evolved together, the mechanisms by which the fungi provide stress tolerance and resistance to its host, and how this association may affect the plant’s rhizosphere.

## 2. Evolution of *Epichloë* Mutualistic Association with Grasses

*Epichloë* fungi often form symbiotic mutualistic relationships with cool-season grasses. Nearly 30% of cool-season grasses across the globe are known to form a relationship with *Epichloë* species [5]. The fungus provides the plant with protection through secondary metabolite production and the plant provides nutrients and accommodation to the fungus. The alkaloids produced by *Epichloë* species have four major classes called lolines, indole-diterpenes, peramines, and ergot alkaloids. These different alkaloids confer protection against a variety of abiotic and biotic stressors. Peramines and lolines are known for their insect deterrence and insecticidal properties, whereas ergot alkaloids and indole diterpenes have mammalian toxicities [6].

*Epichloë* fungi can have sexual and/or asexual transmission. While sexual *Epichloë* mutualism with grasses does occur, such as *E. typhina* with *Dactylis glomerata* (orchard grass), it is more common that asexual forms of the fungus inhabit perennial cool season species [7]. Some common asexual *Epichloë* relationships with cool-season grasses include *E. coenophiala* with *Festuca arundinacea* (tall fescue), *E. festucae var. lolii* with *Lolium perenne* (perennial ryegrass), and *E. occultans* with *Lolium multiflorum* (annual ryegrass). 

Phylogenetic, genetic, and physiological evidence suggest that *Epichloë* endophytes and cool-season grass hosts co-adapted together over a long period of time [8,9]. Most of the mutualistic *Epichloë* species are asexually reproduced by vertical transmission. Evolutionary theory states that vertical transmission is an indicator of a strong mutualist as asexuality leads to very little response to selection due to decreased recombination and genetic diversity [8,10]. This contrasts with the host that reproduces sexually and therefore can quickly respond to selective pressures. This system of the host having increased gene flow compared to the symbiont is consistent with Law’s Hypothesis, which states that loss of sexual reproduction is often a precursor to mutualism, as it reduces the chance of the symbiont becoming pathogenic again. Many asexual *Epichloë* species are interspecific hybrids that arose from sexual progenitors. The exact mechanism by which hybridization occurred is unclear, but it likely arose from the fusion of hyphae and nuclei or abnormal segregation in mating between sympatric *Epichloë* species [11]. The change from sexual reproduction to asexuality eliminates a primary mechanism of pathogenicity [12]. The asexual interspecific hybrid species often show increased fitness compared to their sexual progenitors by alkaloid gene loci pyramiding [13], which likely led to more than half of *Epichloë* species that are now hybrids [12]. Post-hybridization, co-adaptation of the fungus and grass occurs in a host-specific manner. These facts present a strong argument for a long evolutionary history of mutualism between *Epichloë* species and their grass hosts.

It is likely that asexual hybrid *Epichloë* symbionts co-diverged at the same time [8,9]. Previous research using these genes has found that many *Epichloë* species display co-phylogeny with their grass host [14]. Schardl et al. [9] found that early cladogenesis events corresponded between *Pooidae* (a subfamily of *Poaceae*) and *Epichloë*. They also found that within the tribe *Poeae,* a group that contained *Lolium* species showed mirrored topology to their *Lolium*-associated-clade endophytes. These results provide evidence for co-divergence between the species.

## 3. Mechanisms of Interaction between Grass and Endophyte

*Epichloë* species are known for producing bioprotective compounds known as alkaloids [15]. The most commonly produced alkaloids are lolines, indole-diterpenes, ergot alkaloids, and peramines. Genes or gene clusters have been identified for these alkaloids. These gene clusters consist of 10 to 12 genes for EAS (ergot alkaloid), IDT (indole-diterpene) and LOL (loline), depending on the species [16]. A single gene, PER, was found to be responsible for peramine synthesis [17]. The pathways for each alkaloid are complex, with many intermediate metabolites produced that affect alkaloid composition [18]. Endophyte-infected (E + ) grasses have been documented to gain protection from drought, insects, nematodes, cold, flooding, heavy metals, pathogens, and mammalian herbivory [19]. Loline and peramine alkaloids are known insect deterrents and insecticides [20,21]. Ergot alkaloids and indole-diterpenes alkaloids have mammalian toxicity properties as well as some insect protection [13]. E+ plants can withstand drought and edaphic stress at a higher level than uninfected plants (E-) [21]. Research shows that endophytes can interact with the grass host to modify the physiology and biochemistry of the plant, such as rolling and shedding leaves under drought stress [22]. These types of modifications likely translate to other stressors as overall yield and persistence in a variety of environments are higher for E+ grasses than E- grasses [21,23,24]. The exact protections that the endophyte provides depend on alkaloid genes present as well as the genetic interplay between the endophyte and the host. 

The exact mechanisms that lead to the successful mutualism between *Epichloë* species and cool-season grasses are not well defined. Most plants, when faced with a fungal invasion, will activate defense responses such as a hypersensitive response, which triggered cell death. *Epichloë* species do elicit defense responses from non-host plants [25]. However, endophyte-infected host grasses do not produce these responses against the endophyte. The endophyte lives and grows in the intercellular space within the plant, in synchrony with the plant maturation [20]. The fact that the plant does not react to the endophyte as a pathogen, and that the endophyte only grows at the same pace as the grass, suggests a significant crosstalk between the two species. There are a few mechanisms by which this may occur. A recent study proposed that the fungus secretes an inhibitor to papain-like cysteine proteases (PLCPs). PLCPs are a part of the innate plant immune response that acts to trigger signaling cascades when met with pathogens. Some pathogens utilize plant cysteine protease inhibitors to decrease the PLCP response. Passarge et al. [26] found that *E. festucae* acted similarly by reducing four active PLCPs in perennial ryegrass (*L. perenne*). They hypothesized that a yet-to-be-identified fungal secreted effector inhibits PLCPs. Other research has shown that endophyte-infected grasses have higher levels of phenolic compounds and other antioxidants compared to endophyte-free grasses [27]. Other studies have postulated that the grass recognizes the endophyte through increased production of resveratrol and chitinase, as well as other phenolics [28]. Reactive oxygen species (ROS) have been shown to regulate the fungal growth within perennial ryegrass. Tanaka et al. [28] identified a fungal mutant with an insertion in the NADPH oxidase gene noxA, which leads to dramatically increased fungal growth within the plant. NoxA is a hydrolase enzyme that regulates superoxide production. They detected ROS in the non-mutant mutualistic fungus, but not in the *noxA* mutant, leading them to conclude that ROS restricts fungal growth. Further ROS production is regulated by SakA (stress-activated mitogen-activated protein kinase A). It was found that when SakA is deleted, ROS production increases [25]. It is well known that grasses and endophytes can both change the metabolite pathways within themselves to produce different compounds [29], but research has shown that in some cases, the plant and endophyte can work together to produce a metabolite that could not be synthesized by either species alone [30]. There is also evidence that the crosstalk will vary between plant parts. Transcriptome analysis found that fungal gene expression differs between floral and vegetative parts of *Epichloë*-infected grasses [31]. This finding provides further evidence for the complicated nature of this mutualism. 

Other essential factors in the mechanisms of mutualism have been found. Two Rho GTPases, Cdc42 and Rac4 have been implicated in intercalary growth regulation and hyphal network formation [32]. ProA, a transcriptional regulator, appears to work in tandem with symB/symC, membrane-associated proteins, to act on NOX to regulate fungi growth [33]. Additionally, Green et al. [34] found a component of STRIPAK (striatin-interacting phosphatase and kinase) called MobC to be important for mutualism and hyphal cell to cell fusion (Figure 1). 

Specific genes have been found to be essential in the change from pathogenic infection to a mutualistic relationship in the endophyte and the host grasses. Eaton et al. [25] disrupted components of the Nox complex and SakA, previously revealed to be necessary for mutualism, in *L. perenne* infected with *E. festucae* and determined the transcriptomic changes that occurred (Figure 1). They found that certain classes of genes were up or down-regulated in the grass, most notably genes involved in pathogen defense, transposon activation, hormone biosynthesis, and response [25]. Further work carried out by Eaton et al. [35] identified 182 specific fungal genes associated with mutualism in the *L. perenne* and *E. festucae* system. They identified the upregulation of genes that encode degradative enzymes, transporters, and primary metabolism and the downregulation of genes that encode for secondary metabolism and production of small-secreted proteins. These are genes involved with nutrient-starvation response and indicate that disruption of the mutualism results in the endophyte, taking in more nutrients from the host, including cell wall components, consistent with pathogen feeding patterns. Further evidence of transcriptome changes that aid in mutualism has been found. Schmid et al. [36] discovered gene expression changes in fungi in growing grass compared to those in mature grasses with genes shifting from hyphal growth to alkaloid production [36]. New genes important for symbiosis continue to be found. Hettiarachchige et al. [17] recently identified novel genes at different stages of seedling development with *Epichloë* infection. They found that the first 0–4 h of growth showed the highest amount of differentially expressed genes, suggesting that genetic cascades for symbiosis are triggered quickly. Additionally, they found different novel candidate genes for secondary metabolites between the different strains of endophyte studied. This provides evidence of the highly specific nature of the endophyte-grass association. 

## 4. Environmental Effect on Endophyte/Grass Associations 

The environment can play a substantial role in *Epichloë* endophyte infection and production within the grass hosts. Temperature, season, humidity, climate, soil nutrients, etc., have been shown to have an effect on endophytes. Alkaloid concentrations can be different in the same grass/endophyte combination in different geographical locations [37]. It is known that endophytes tend to be more prevalent in the plant in summer than in winter. Seeds exposed to hot, humid weather for an extended period can lose their endophyte. It was found in growth chamber studies that very cold weather may affect fungal transmission to the grass progeny [20]. Temperature can also influence individual alkaloid production [38] and soil nutrients can influence endophyte infection. Rasmussen et al. [39] found that *L. perenne* plants had lower levels of endophyte concentration and alkaloid production in soils that were provided with high levels of nitrogen and carbohydrates. One may think that increased nutrients to the grass would only increase the fungal mutualist. However, research investigating the environmental effects on the *Epichloë*-grass relationship in natural and agronomic systems has mixed results. Every host of a mutualistic relationship is performing a cost-benefit analysis to ensure that the cost of the endophyte does not exceed the gain the host obtains [40]. In stressful environmental conditions, the E+ plants persist better than E- plants (Figure 2). This is the basis of the agronomic use of these grass-endophyte associations. However, reports of the endophyte over-taxing the host in low-input situations have been reported in some grasses leading to decreased yield [41,42]. It appears that in some situations, the endophyte does tax the plant more heavily, but the benefits of the fungus still outweigh the costs. Bacon [43] found that while E+ seeds needed more water to germinate and E+ seedlings required more nutrients, the overall E+ grasses were more tolerant in conditions of abiotic stress. These findings highlight the importance of the specific endophyte–host relationship. Moreover, it is possible that different environments can change the effect of the same host–symbiont relationship due to phenotypic plasticity and potential epigenetic changes [41]. How the environment changes mechanisms by which endophytes and grasses interact is not well understood. 

## 5. Stress Tolerance Mechanisms

### 5.1. Drought

Several studies cite the connection between *Epichloë*-infected grass and drought tolerance. Three main mechanisms are thought to be the reason for this: increased root biomass [44,45,46], better regulation of stomatal closure, and higher accumulation of solutes for osmotic regulation [47]. Xu et al. [48] investigated *E. sinensis* association with *Festuca sinensis* under different soil water contents and found that the E+ had better growth in roots and shoots, photosynthetic rate, ion accumulation, and nutrient accumulation. Additionally, they looked at phytohormones and found that the E+ had higher levels of abscisic acid (ABA) under drought conditions compared to the E- grass. ABA is important for drought response and functions to trigger stomatal closure and induce the production of protective compounds [49]. A recent study investigated other phytohormones salicylic acid (SA) and jasmonic acid (JA) of E+ and E- drunken horse grass (*A. inebrians*) at different levels of soil moisture [49]. They found that the E+ plants had overall lower levels of SA and JA compared to the E- grass at all soil levels, but that SA increased with soil moisture and the lowest recorded JA amount had a relatively high soil moisture content. Overall, the E+ plants performed better under the low soil moisture. It is not clear how these plant hormones play into drought resistance yet, but the endophyte appears to modulate them to some degree (Table 1). Other studies have found higher levels of sugars, proline, and glutamic acid found in drought-stressed E+ tall fescue, as well as higher levels of mannitol and loline alkaloids [50]. These are fungal-derived metabolites that could act as osmotic regulators. Gene expression relating to drought tolerance was found to be altered in perennial ryegrass infected with *Epichloë festucae.* Dupont et al. [51] showed that drought-related genes were downregulated in E+ perennial ryegrass, decreasing the grass’s drought sensitivity. The endophyte likely makes up for this reduced sensitivity by producing the osmotic protective solutes. Endophyte infection also increases gene expression of galactinol synthases in the grass, which produces osmo-protective oligosaccharides [51]. The endophyte promotes the production of antioxidant enzymes such as superoxide dismutase, ascorbate peroxidase, catalase, and guaiacol peroxidase under heat and drought stress, which mitigates damage done by the accumulation of ROS (Table 1) [52].

### 5.2. Salinity

Studies have shown that grasses infected with an *Epichloë* endophyte outperform uninfected plants in situations of salt stress [53,54,55]. Na^+^ is an important regulatory element for ion content in cells but can become toxic at high levels. The endophyte appears to regulate ion concentration and help with nutrient absorption as ways to counteract Na+ stress [55]. Reza Sabzalian et al. [53] found that when presented with salt stress, there was less Cl^-^ and Na^+^ in E+ *Festucae arundinacea* roots and an increase in K^+^ in the shoots, compared to the uninfected fescues. Creating a higher ratio of K^+^ to Na^+^ could help balance ion content. Other studies show increases in P and N, in addition to K^+^ as ionic adjustment ameliorations [55]. In addition to regulating ionic elements, the endophyte may also increase net photosynthesis, which is often lowered under abiotic stress [56]. Much as with drought stress, the osmo-protective solutes that endophytes produce also play a function in salt stress. Notably, glycinebetaine (GB) has been found to be higher in *Epichloë* infected sorghum and is good at mitigating oxidative stress [57,58]. Increases in other enzymatic and non-enzymatic antioxidants have been found that help keep ROS from accumulating under salt stress [57,58]. *Epichloë* endophytes also change the anatomy of the host plant to combat salt stress. Chen et al. [59] found that wild barley infected with *E. bromicola* had increased xylem, phloem, and vascular bundle area compared to E- plants. Vascular tissues tend to decrease dramatically with salt stress. They also found that the E+ wild barley had thicker leaf veins, stem epidermis, cortex, and endodermis (Table 1). These anatomical changes keep water loss low and conduction high, which helps maintain the plant’s essential physiological functions.

### 5.3. Heavy Metals

Heavy metals in the soils can become toxic to plants and interfere with biological processes. There is evidence that endophyte infection can alleviate symptoms of heavy metal stress. Zhang et al. [60] found that drunken horse grass infected with *E. gansuensis* had more biomass and tillers and was taller in height than uninfected grass under cadmium stress. Similarly, studies have found that E+ perennial ryegrass shoot biomass was better in E+ under cadmium stress and had increased amounts of H_2_O and chlorophylls a and b [61,62]. A recent study found that E+ perennial ryegrass accumulated copper and cadmium better than its E- counterparts [62]. When presented with Zn stress, E+ perennial ryegrass showed less Zn accumulation in the leaves and had a higher tiller number and overall biomass. The overall better performance is likely due to the lack of Zn in the leaves, where it could interfere with photosynthetic processes [63]. Antioxidants are important to increase osmotic regulation and photosynthesis, and to counteract the ROS accumulation associated with heavy metals. A study carried out by Salehi et al. [64] found mixed results in the performance of different populations of E+ perennial ryegrass, but overall they found increased antioxidant enzymes in E+ grass. They also found an increase in the expression of LpTIP1;1, an aquaporin gene. This may cause changes to water content in E+ grasses. In general, *Epichloë* endophytes likely ameliorate the effects of heavy metals by increasing antioxidant enzyme activity, as well as maintaining plant growth (Table 1).

### 5.4. Low Nutrients

A variety of studies have shown *Epichloë* infected grasses could be more tolerant to low nutrient inputs [65,66,67,68,69]. Wang et al. [65] studied the mechanisms by which E+ drunken horse grass was more tolerant to low N environments. N is important to study as it is often the first limiting nutrient in major crops [69]. They found that the endophyte regulates G6DPH, a pentose phosphate pathway enzyme associated with abiotic stress tolerance, as well as glutathione activity, NADPH ratio, and photosynthesis. The same group also studied N use efficiency in the same grass/endophyte association. They found an increase in N-related enzyme activity, including an increase in nitrate reductase leading to increased NO3- and an increase in nitrite reductase leading to increased NH4+. The creation of these substrates is vital for N use in plants. Glutamine synthesis was also increased in E+ plants, which is essential in converting inorganic N to amino acids [65]. A more recent study echoed the finding that G6DPH regulation is involved in low N environments [69]. The same group also found that *E. gansuensis* reprogrammed the metabolism of drunken horse grass to increase organic, fatty and amino acids in the plant (Table 1). Organic acids are secreted when the plant is faced with nutrient stress and promotes the uptake of minerals from the soil, therefore an increase in them would be beneficial to the plant under stress. 

Phosphorus is another nutrient essential for plant growth and a deficiency in P can lower the bioavailability of other nutrients. A study conducted on drunken horse grass and *E. gansuensis* found that the E+ grass was more tolerant to low P than the E- grass. This is likely due to the endophyte regulation of certain amino acids, amino acid metabolism and organic acids that were found in E+ plants, which helps with nutrient metabolism and photosynthesis [68]. Additionally, an increase in vanillin, a ROS scavenging compound, which helps decrease oxidative damage, was found in E+ grass (Table 1) [68]. Nutrient cycling in endophyte-infected plants is commonly tied with changes in the surrounding microbiome. The effects of the endophyte on vital microorganisms will be explored in a later section. 

### 5.5. Cold

The effect of endophyte infection in grasses for cold tolerance is mixed. Heineck et al. [70] found no differences in freezing tolerance in E+ and E- perennial ryegrass. However, Chen et al. [71] found that E+ drunken horse grass had higher seed germination rates compared to E- grass in cold conditions. This group analyzed gene expression changes in the E+ grass and found 152 differentially regulated genes, including genes involved in fatty acid biosynthesis, stress response, and protein turnover (Table 1). Upregulation of unsaturated fatty acids was found, which is consistent with cold tolerance [72]. It is interesting to note that ergot alkaloid compounds were accumulated at higher levels in cold stressed *Epichloë* infected *Festuca sinensis*, indicating that the cold has some effect on the fungi alkaloid content [73]. 

### 5.6. Flood

Much like with cold stress, the effects of the *Epichloë* endophyte on waterlogging stress are mixed. Arachevaleta et al. [74] initially found that there was no difference in E+ and E- tall fescue under water stress. Similarly, Adams et al. [75] investigated two types of bluegrasses, one that lives in water logged areas and one that lives in drier environments. They found that E+ infection did not change performance in flood conditions. However, Song et al. [55] found that the endophyte-infected wild barley had increased tolerance to flooding. They found that the endophyte increased production of proline (an osmotic regulator), had decreased electrolyte leakage and decreased malondialdehyde content, a lipid peroxidase damage indicator. These results suggest the endophyte aids in osmoregulatory processes and protection from oxidative damage (Table 1) [76]. More recently, Saedi et al. [77] tested two strains of *Epichloë* infected tall fescue in waterlogged, anaerobic conditions. They also found mixed results with one E+ tall fescue type performing well and the other E+ tall fescue type performing poorly. They hypothesized that the poor performing tall fescue was due to endophyte incompatibility. The E+ tall fescue that performed well had increased biomass, increased osmoregulation, and decreased catalase and ascorbate peroxidase compared to E- controls. This is consistent with the conclusion that the endophyte aids in oxidative damage protection and osmoregulation. Wang et al. [78] conducted a waterlogging study on *F. sinensis* to investigate biomass differences in E+ and E- grass. They found that while all plants suffered under flood conditions, the E+ plants had higher root to shoot ratios. This could be a result of increased photosynthesis, even under stress (Table 1). 

### 5.7. Pathogens

Disease resistance is key in any crop production. Diseases can arise from bacteria, fungi, viruses, parasites, or other microorganisms. The ability of *Epichloë* fungi to protect the host species from a variety of fungal pathogens is well documented [79]. The mechanisms by which fungal endophytes can protect the host plant against disease are numerous. For fungal pathogens, one of the most apparent mechanisms is that the endophyte occupies the same ecological niche that the pathogen would inside the plant [80]. Some *Epichloë* endophytes, such as *E. festucae* and *E. inebrians*, produce a secreted antifungal protein, Efe-AfpA, that directly impedes fungal pathogens [81]. More generally, endophytes can ease disease damage by increased production and activity of antioxidants. ROS can accumulate under pathogen stress and cause damage. Endophytes enhance the production of compounds like superoxide dismutase and peroxidases that scavenge for ROS. Studies have also found an increase in proline, another ROS scavenger, and a decrease in malondialdehyde in E+ perennial ryegrass under pathogen stress compared to E- perennial ryegrass (Table 1) [82,83,84,85,86,87]. A recent study found that when spray was inoculated with ergot, drunken horse grass infected with *E. gansuensis* combated the disease successfully though increased antioxidant enzymes superoxide dismutase, peroxidase, catalase and proline and decreased malondialdehyde content [88].

Salicylic acid (SA) is essential in plant pathogen defense. Once triggered, SA will lead to the activation of resistance genes which will then make pathogenesis-related proteins. *Epichloë* endophytes are likely pathogens turned mutualists and therefore their presence could make the plant host more primed for a pathogen response. In fact, Wang et al. [89] found that Chinese wild rye that was E+ produced more SA when presented with two different pathogens as compared to the E- grass. They also found increased lignin content, which has been shown to be protective to pathogens [90]. However, some research has found downregulation of SA-related genes [51]. Suppression of SA pathways usually enhance jasmonic acid (JA) pathways. Further evidence for enhanced JA pathways is the upregulation of gene *TF441*, which may produce precursors to JA, but the effects of this are unverified [91,92]. Other studies have found that the endophyte upregulates SA production through increasing expression in genes involved in SA synthesis and response when challenged with a pathogen [93]. These studies show that the endophyte may upregulate or downregulate SA and JA depending on the exact interactions at play. The quinate and shikimate pathways are important in pathogen defense. Rasmussen et al. [94] found increased amounts of both compounds, as well as other phenols, in E+ perennial ryegrass. Other phenolic compounds are known to be important in plant defense and are part of the bioprotective phenylpropanoid pathway [95]. Dupont et al. [51] found upregulation in genes encoding for enzymes involved in phenylpropanoid production. Other studies have shown that endophyte infection not only increases in phenolic compounds but also volatile organic compounds (VOCs). Some VOCs have been shown to inhibit fungal growth [96]. Zhang et al. [97] found that volatile oils extracted from E+ drunken horse grass significantly reduced fungal growth compared to E- grass (Table 1). VOCs can also function in plant stress response communication [98]. It is well described that certain *Epichloë*-produced alkaloids peramine and loline have anti-insect properties that could deter infection. Endophytes reduce viral infection in host plants, namely by reducing insect vectors such as aphids. The insect deterrence mechanisms will be explored in another section [99]. Overall, the endophyte can increase resistance to diseases through alkaloids and other compounds, while decreasing the trade-off that normally occurs between plant growth and plant defense [100].

### 5.8. Nematodes

It is well known that wild-type endophyte-infected grasses have a negative effect on a variety of nematodes [101,102,103]. Various studies showed the ability of *Epichloë* infected perennial ryegrass and tall fescue to be resistant to endoparasitic nematodes, such as root-knot nematodes and *Pratylenchus scribneri* [104]. However, ectoparasitic nematodes do not appear to be affected. Kimmons et al. [101] found that endophyte-infected tall fescue was successful in controlling *P. scribneri* and *Meloidogyne naasi* but had very little effect on the ectoparasitic *Helicotylenchus pseudorobustus.* A more robust study from Guo et al. [105] tested the effects of E+ plant-soil on 11,889 soil nematodes from 37 different genera. They found significant differences between plant-soil types with the plant-parasitic index lower in E+ plant-soil than E- plant soil. This is interesting because endophytes do not infect the below-ground tissue of the grass.

Nematode control may be linked to one or more of the main alkaloids produced by the endophyte. It is thought that the ergot alkaloid is the compound toxic to nematodes. This is the same alkaloid that is toxic to mammals. It is interesting to note that peramine is the alkaloid that confers resistance to insects, but strains of endophyte that contained peramines but no ergot alkaloids did not confer high levels of resistance to *Pratylenchus* spp. [106]. In fact, these strains had no better effect on controlling nematodes than endophyte-free strains of tall fescue. A later study found that the ergot alkaloid compound ergotamine was found to control *P. scribneri* in vitro [107]. However, a study carried out with perennial ryegrass found that the grass lacking ergot alkaloids but containing other alkaloids was still able to control *P. scribneri* just as well as the grass that did have the ergot alkaloids [108]. There is no published concrete evidence for the mechanisms of nematode control, but there are a few theories. One is that the endophyte changes root exudates in some way that is detrimental to nematodes. It is also possible that morphological changes in the plant, like epidermal thickening, could lead to decreases in nematodes (Table 1). As far as nematode tolerance in E+ grasses is concerned, this could also be due to the other benefits that the endophyte confers that overall makes the grass healthier. 

### 5.9. Insects 

Insect tolerance and avoidance is an area that has been heavily studied in *Epichloë* infected grass species. There are at least 45 different insect species that have been found to be negatively affected by *Epichloë* alkaloids [109]. Insect resistance is mainly linked to peramine and loline alkaloids, but indole-diterpene alkaloids have also shown insecticidal properties [27,110]. N-acetylnorloline and N-formyloline are loline alkaloids that have been shown to be toxic to different insects [111]. Bastias et al. [92] found that lolines likely have a profound effect on aphid metabolism. The energetic cost of detoxification leads to decreases in development and reproduction [112]. Expoxy-janthrims are the main indole-diterpene alkaloid that confers insect toxicity; however, nodulisporic acids are also indole-diterpene metabolites that were shown to be toxic to certain insects by binding to arthropod-specific glutamate-gated chloride channels [36,113]. Insect resistance can be further heightened by the grass–fungus interaction. Studies have shown that clipping leaf tissue can lead to increased production of lolines [114]. Much like in pathogen defense, plants utilize SA and JA systems for insect defense. JA pathways may be especially heightened in chewing and biting insects. The same mechanisms of endophyte infection that enhance these pathways in pathogens apply to insects. Some defense mechanisms may slow down plant growth that endophyte infection can override. He et al. [115] studied endophyte response to aphids and found that the endophyte increased plant growth by decreasing SA content and increasing JA. When aphids attack, the plant responds with SA and phenylalanine ammonia-lyase, which slows down plant growth. The transcription factor WRKY54 was found in E+ plants to combat the slowed-down plant growth. The endophyte also reduced DghWRKY57, which leads to increased JA, an antagonist to SA (Table 1) [116]. Incredibly, there is some research indicating that *Epichloë* endophytes can lead to multi-trophic interactions that benefit the plant. Fuchs et al. [117] found that E+ plants attracted more hoverfly pupae and larvae. Hoverfly (*Syrphidae* family) is a predator of aphids, therefore decreasing aphid infestation. They propose that volatile compounds may change phytohormones produced that are to do with olfactory attraction. Hennessy et al. [118] recently conducted a study and found that despite using olfaction to orient to perennial ryegrass and different volatile compositions between E+ and E- plants, Argentine Stem Weevils (*Listronotus bonariensis*) could not distinguish between the infected and uninfected plants on olfactory senses alone. This may not be the case with other insect species. There is evidence that African black beetles (*Heteronychus arator*), as well as the grub *Costelytra zealandica*, avoid E+ plant material based off olfaction (Table 1) [119,120]. 

### 5.10. Weed Competition

*Epichloë*-infected grasses tend to do well in weed competition. This may be the indirect effect of the plant being overall healthier due to the endophyte. Some endophyte-infected grasses may take weed resistance further with allelopathy. Allelopathy is the ability of one plant to suppress another plant via chemical exudation. Some plants can release chemicals that prevent seed germination in other plant species or modify the microbiome to negatively affect other plant species. There is evidence that E+ perennial ryegrass can suppress white clover with allelopathy playing a role [121,122]. Tall fescue has been reported to cause an allelopathic effect on various species including birds foot trefoil, red clover, and white clover [123]. More recently, Vázquez-de-Aldana et al. [124] found that red fescue infected with *E. festucae* increased the allelopathic potential of the red fescue on red and white clover. The endophyte may not be directly responsible for these allelopathic interactions as grasses can produce allelopathic chemicals, such as m-tyrosine, on their own [124]. However, it is known that endophytes that live in the above-ground parts of the plant can cause increased exudation of phenolic compounds and other metabolites from the roots, including syringic acid and myristic acid [125,126]. These acids have been shown to suppress certain weeds [127,128]. Phenolic compounds, like syringic acid, can interfere with enzymes that are important for the physiological processes of the plant, such as mineral uptake and photosynthesis. Myristic acid is a lipid that also has a negative effect on plant protective enzymes (Table 1) [129]. The prevention of activation of these important enzymes could inhibit or block plant growth.

### 5.11. Animal Herbivory

There is quite a bit of literature on the physiological effects that the endophyte-produced alkaloids can have on livestock [130,131,132,133]. Ergot alkaloids are the primary class of alkaloids that cause problems within the United States cattle industry, causing disease in E+ tall fescue grazing cattle called fescue toxicosis. Indole-diterpenes can cause negative effects in livestock like sheep and affect countries like New Zealand where sheep make up a large sector of their agricultural economy. Indole-diterpenes are known for causing a neurological condition called “ryegrass staggers” in sheep grazing perennial ryegrass (*L. perenne*).

Ergot alkaloids and indole-diterpenes create a neurological effect on mammals. Ergot alkaloids consist of clavines, lysergic acid and its derivatives, and ergopeptines with lysergic acid and ergopeptines sharing the same pharmacophore. This pharmacophore (D-lysergic acid) is structurally shaped very similarly to neurotransmitters like serotonin. Different alkaloid metabolites are thought to affect different physiological processes within certain mammals. For example, Trotta et al. [134] found that ergopeptine can influence vasoconstriction by binding to serotonin receptors found in the mesenteric blood vessels (Table 1). Similarly, Zhang et al. [135] discovered that ergonovine has a cytotoxic effect on the smooth muscle cells of animals. Some more recent studies by Mote et al. [136], Mote et al. [137], Mote et al. [138] provide evidence that ergot alkaloid metabolites greatly affect the metabolome and gut microbiome of cattle and leads to toxicosis. It was found that grazing E+ tall fescue led to increases of *Ruminococcaceae* and *Lachnospiraceae* families of bacteria. Both families are associated with decreased weight gain and increased respiration in cattle. It was also found that E+ tall fescue grazing led to increased temperature sensitivity of the fecal microbiota and caused metabolic changes associated with inflammation. The microbiota changes correlated with the metabolomic changes in response to temperature. This provides evidence that the gut microbiome and metabolome may play a more important role in fescue toxicosis and offer new insights for intervention (Table 1) [139]. 

Non-livestock animal herbivores are also negatively affected by endophytes. A study carried out by Rudgers et al. [140] found that voles (*Microtus* spp.) would avoid E+ tall fescue in favor of other surrounding vegetation. This was not the case in E- tall fescue plots. Conover et al. [141] found that Canada geese (*Branta canadensis*) lost body mass when feeding on E+ tall fescue compared to those feeding on E- tall fescue. Likely the same mechanisms by which alkaloids impact livestock also impact non-livestock animals.

## 6. *Epichloë* Infected Grass Interaction with Rhizosphere Microbiome 

The rhizosphere is a small yet dynamic zone of biological activities and interactions between plant roots and soil microbiome [142]. A robust and diverse rhizosphere is a strong indicator of soil and plant health [143]. In terrestrial ecosystems, nutrient cycling and transfer of energy fluxes are closely tied with organic matter composition, a mixture of root hairs and exudates, active and decayed root cells, volatile organics, and proteins. These energy substrates are collectively called rhizodeposits [144]. While these rhizodeposits have a plant origin, soil microbes often contribute to the breakdown of nutrients that help allow for plant uptake and can influence plant growth and physiology [145]. Therefore, plants form a plant-microbe feedback loop with their rhizobiome which, in turn, affects soil and microbiome composition. Thus, a comprehensive understanding of belowground root functionality and soil composition is required to explore the prospects of microbial symbiosis and its role on soil and grass species health [146,147,148]. The mechanisms of species coexistence are important in understanding and predicting species diversity aboveground and belowground [149]. Species coexist mainly by niche partitioning [150] (diversity in habitat, nutrient use, and survival strategy), especially in microbial communities, where they self-limit themselves as their useful resources decline. However, for sedentary species such as plants, the greatest challenge in realizing their coexistence mechanism is their high resource overlap with each other [150,151,152]. Recent studies have indicated that plant-associated microbes could initiate and establish “self-limitation” for plants [153]. This ‘extended plant phenotype’ that includes plant-associated microbiota is critically important for plant productivity and survival where resources are transferred and allocated according to resource requirement [153,154]. The rhizobiome composition, diversity and functionality, as well as the composition of rhizodeposits, may experience changes depending on region, climatic variation, soil type, plant diversity and the presence of biotic and abiotic stressors [155,156,157]. Two main soil microorganisms need to be considered when analyzing the plant–microbiome relationship. (i) Bacteria: often considered the most important soil microbe. They are essential for decomposing residues and nutrient cycling. They especially help increase N and P availability to the plant [158]. Actinomycetes, a lignin-degrading gram-positive mycelial bacterium is worth mentioning in this case because often they are classified on their own. (ii) Fungi: mycorrhizal fungi establish symbiotic relationships with plants and help increase water and nutrient uptake and can help stabilize soil aggregates [159]. In legumes, mycorrhizal fungi also help improve N fixation. Studying and manipulating the rhizobiome of plants is becoming increasingly popular for productivity, especially under abiotic and biotic stress [160,161]. 

### 6.1. *Epichloë* Effect on the Rhizosphere

Endophyte infection influences the grass rhizodeposition and therefore an impact on the microbiome. Rhizodeposition can change for several reasons, but one aspect that may influence infected grass’s rhizodeposits is the photosynthetic rate. Grasses infected with endophytes generally have higher photosynthetic rates, which help increase yield [162]. Another way that infected *F. arundinacea*’s rhizodeposition may be altered is by the secretion of alkaloids. Given that the alkaloids are produced by endophytes that live inside the plant, it is not likely that they leach out into the soil via root exudates. However, one study did find a significant amount of alkaloids in the surface soils of an endophyte-infected tall fescue pasture [163]. These alkaloids could largely be the result of E+ tall fescue litter decomposition leaching the alkaloids into the soil. Omacini et al. [164] found that *Lolium multiflorum* infected with *Epichloë* endophyte decomposed more slowly than E- grass but will increase the speed of decomposition of different plant species nearby. This suggests a decrease in microbial activity on the E+ plant material [164]. 

Some specific differences in E+ tall fescue rhizodeposits and E- tall fescue rhizodeposits have been found. One study saw an increase in carbohydrates and organic carbon in the E+ rhizodeposits than in the E- rhizodeposits [165]. Another study found higher amounts of phenolics in E+ root exudates. This could be due to greater phosphorus uptake by E+ tall fescue than E- tall fescue [166]. It must also be noted that the grass species and genotype can have a large impact on rhizodeposition [167]. Studies looking at the effect of grass endophytes on soil composition were the first to suggest that the soil microbiome was impacted by the endophyte. Studies dating back to the 1990s identified that tall fescue endophyte infection had an effect on levels of C and N in the soil [168]. Endophyte-infected tall fescue (E+) have been shown to have differing effects on soil temperature, structure, nutrient levels, etc. Hosseini et al. [169] looked at the endophyte effect on soil structure stability using high energy moisture characteristics (HEMC) for quantification. They found that E+ tall fescue was correlated with increased soil organic carbon, increased stability ratio, and increased stable macropores [169]. Soil organic matter and hot-water soil carbohydrates were also increased, which improves aggregate stability. These changes are possibly due to the endophytic fungus in the tall fescue plant changing the chemical makeup of its root exudates. This is interesting since the endophyte in the shoots must have such internal effects in the plant to act on the roots to the point that the physical qualities of the surrounding soil are impacted. Another study sought to identify the soil C fractions in infected tall fescue soil and uninfected tall fescue soil. Since there has been evidence that infected tall fescue leads to higher levels of C sequestration, it was hypothesized that E+ tall fescue would have higher C fractions. C sequestration is the ability of the soil to fix atmospheric C. This provides important environmental effects, as well as providing an essential nutrient to the plant and soil microorganisms. Handayani et al. [170] looked at C levels in the two tall fescue systems over the course of four years. They took data on microbial biomass, particulate organic matter C, microbial biomass C, mineralizable C, and C in micro and macro aggregates. They found that microbial biomass C and mineralizable C were the most sensitive indicators. Microbial biomass C (C contained within the microbes in the soil) was higher in E+ fields in shallow soil (0–15 cm) but did not differ at deeper depths. Mineralizable C (C available to the plant) was lower in E+ fields at both depths. These results suggest that the C in the E+ soil was less labile than the C in the E- soil. Micro-aggregates showed less C in infected tall fescue soil than in the uninfected field. 

Leaf material in the soil may have a differing effect on the soil microbiome than living roots. Franzluebbers et al. [163] conducted a study that looked at the effect of leaf material in the soil in the long term and in the short term. They found that in the period of 32 days E+, the material resulted in decreased C mineralization and a decrease in microbial biomass C. This is consistent with studies conducted with living plants. However, they did find that mineralizable N increased, as well as microbial biomass N in the E+ leaf material soil (Table 2). These changes may be due to the phenolic compounds that are released from the leaves as they decompose [171]. In the longer-term study, they found an accumulation of C and N in the soil. Slower decomposition of litter can lead to C and N accumulation in the soil over time. When evaluating potential mineralizable C and N in the long term, they found a 5% decrease in mineralizable C in the infected leaf tissue soil (Table 2). However, there was no change for potentially mineralizable N. Measurements of ergot alkaloid levels in both E- and E+ soils were taken, and it was found that the soil that had E+ tall fescue growth for many years had 2.7 times more ergot alkaloid than the E- plots. The release and accumulation of the various compounds that are released from infected tall fescue as it decays influences the soil microbiome and the nutrients in the soil. These effects could be long-lasting or short-lived. The mechanisms underlying how endophytes influence these processes, how this changes the access to sparingly soluble soil P, and how it might vary with endophyte strain or differences in soil P speciation and availability are still not known. Insight into the soil composition under E+-infected grasses can provide a starting point for evaluating microbial changes.

### 6.2. Microbiome Changes in Response to Stress

As discussed above, the rhizosphere of a plant may change under abiotic stress conditions such as drought, flood, and salt pressure. The way that the microbiome changes will vary significantly from one grass to another, as well as from one environment to another. For instance, drought stress has been found to negatively impact almost every part of the plant-rhizosphere ecosystem. This is expected as a decrease in moisture can lead to cell lysis and death in microbes. Additionally, under drought stress, plants will decrease root biomass development and exudation, which will lead to less nutrient supply for the microbiome [172]. Naylor et al. [173] investigated the root microbiome reaction to drought stress across 18 different grass species. They used 16S rRNA gene sequencing and determined that microbial shifts under drought conditions were conserved across all species studied with actinobacteria being enriched. This enrichment could be due to cell wall component changes [173]. Furthermore, this drought tolerance in plants has strongly been linked to ‘symbiogenics’ or ‘symbiosis-altered genetic expression’, where certain microbial species colonize plant roots and confer drought resistance [174]. 

Nutrient stress is another abiotic stressor that may negatively impact the microbes that live in the soil. Generally, under decreased nutrient intake, plants will have fewer root exudates released and therefore the microbiome will have fewer nutrients to scavenge as well [175]. While there is very little research that has studied changes in grass microbiomes under abiotic stressors, it has been documented that a robust microbial community is associated with abiotic stress tolerance in a variety of crop species [176]. Likely, this is the case for grass species as well. For example, under most stressors, plants will produce ethylene, which will eventually lead to senescence. The presence of any 1-aminocyclopropane-1-carboxylic acid (ACC) deaminase-producing bacteria will decrease the ethylene levels and therefore decrease damage to the plant [177]. 

Much like with abiotic stresses, microbe response to biotic stresses is mixed. Once again, little research has been done with grass species, but other studies in different plant species have found some interesting results. In cotton that was diseased with *Verticillium*, it was found that beneficial bacteria and fungi were greatly reduced and saprophytic fungi were increased [178]. Similarly, aphid infestation of the aerial parts of pepper plants led to root exudation composition changes to recruit bacteria that aided in aphid susceptibility [179]. Once again, enrichment with certain microbes can enhance plant resistance to pathogens, but this may not occur under natural systems. In another study, a microbial-rich compost applied to tomato plants increased resistance to vascular wilt pathogens [180]. In a recent study, improved and balanced macro (calcium, magnesium) and micro (zinc)-nutrient content in the plant has also been attributed to the microbial population (applied as bio-inoculum) in iron-rich soil [181]. Furthermore, a complex trophic level dynamic was reported at 0–5 cm soil depth, where soil native microbes and applied exogenous microbes co-inhabited and improved soil nutrient cycling [182]. In soil, phosphorus is one of the major limited nutrients due to fixation in organo-mineral complex (Fe and Al complexes in acid soil and Ca complex in alkaline soil), thus severely limiting the bioavailability of soil P in soil solution for plant roots [183,184,185,186]. Soils with a low nutrient content are characterized by lower soil microbial biomass and slower soil enzymatic release due to scant soil substrate [187]. Soil P release is a combined action of plant roots (changes in root architecture) and soil microorganisms (mycorrhizal activity, enzyme secretion and organic acid production) [126,188]. 

### 6.3. Microbiome Changes Due to Endophyte Infection

It is well established that *Epichloë* endophyte has some effect on soil nutrients and structure that may affect the bacterial makeup of the surrounding soil. There has been some evidence that alkaloids produced by endophytes may positively affect bacteria present in the phyllosphere. Roberts et al. [189] found that loline alkaloids in fescue encourage the selection of epiphytic bacteria in the aerial parts of the plant. This suggests that the different alkaloids may also enrich the bacteria in the soil. A follow-up study conducted by Roberts et al. [190] looked at the loline effect in the soil microbiome of infected tall fescue. They found that the loline-catabolizing bacteria significantly increased in just one week in the E+ tall fescue microbiome as opposed to the non-loline catabolizing bacteria. The loline-catabolizing bacteria did not increase significantly in the E- tall fescue. Interestingly, they also found that the E+ tall fescue had a greater diversity within the microbiome overall. Shannon’s diversity index numbers revealed that the E+ tall fescue had an H’ value of 4.02 compared to the E- tall fescue’s H’ value of 3.07, indicating greater bacterial diversity. This study provides evidence that E+ tall fescue could positively impact the bacteria makeup of the microbiome in its soil [190]. Another recent study also found a higher Shannon’s Diversity Index in the microbiome of *E. ganseunsis*-infected drunken horse grass (*A. inebrians*) over three seasons [191]. This led to better nutrient acquisition for the plant. However, there has also been evidence that *Epichloë*-infected tall fescue consistently measures less qCO2, which is the rate of mineralizable C per microbial biomass C. This suggests that there is a lower diversity of microorganisms that results in a lower soil metabolism [192].

Further investigation into tall fescue’s endophyte effect on bacteria in the makeup in the microbiome has yielded some interesting results. One study looked at the difference in makeup between Eubacteria subdivisions including alpha, beta, gamma, delta, high G + C gram-positive bacteria, Cytophaga-Flavobacteria and Planctomycetes. It was found that in clay loam soil, E+ tall fescue had a suppressing effect on archaea, high G + C gram-positive bacteria, delta-proteobacteria, and Planctomycetes [193]. Despite these decreases in soil bacteria, it was found that soil C did not change compared to E- tall fescue populations in the short-term, suggesting that the effect was not powerful enough to disturb soil nutrient cycling. Gram-positive bacteria were to be reduced by E+ tall fescue in several experiments. Buyer et al. [194] found that gram-positive bacteria were reduced in E+ tall fescue soil over 60 weeks. Gram-positive bacteria contain common genera of Cocci and Bacilli. Gram-positive bacteria are commonly found in the soil and can serve many purposes. While they can be pathogenic, there is also evidence of them serving as PGPBs [195]. Mahmud et al. [196] found that there were high levels of planctomycetes, proteobacteria, and acidobacteria in E+ tall fescue soil. The proteobacteria and acidobacteria ratio (P/A) indicate soil nutrient richness with a high P/A, indicating rich soil. E- soil had a higher P/A ratio and E+ had a lower ratio. Despite this, the E+ tall fescue performed better than the E- tall fescue, suggesting the endophyte makes up for any nutrient loss. Indeed, it was found that the E+ grasses had a higher available P in the soil. However, other research with E+ drunken horse grass, perennial ryegrass, and wild barley (*Hordeum spontaneum*) has found higher P in E- grass soils, leading to the conclusion that P soil availability may depend on the environmental factors as well as a specific endophyte–grass relationship [197]. The same study found that ammonia and nitrogen oxidizing bacteria were higher in all E+ situations over 18 months. Similarly, Chen et al. [198] found that incorporation of *Epichloë*-infected perennial ryegrass litter into soil led to an increase in ammonia-oxidizing bacteria across different time points. This is consistent with previous studies finding increased soil N in E+ grass soils.

Bacterial makeup may change under stress conditions in E+ soil as well. One study found that the root and rhizosphere bacterial community diversity decreased under drought conditions in *E. gansuensis*-infected drunken horse grass soil [199]. This is in addition to the already lowered root colonizing bacterial diversity seen in the E+ soil. More research has been done on manipulating the bacterial community under stress for plant productivity improvement. Endophyte infected and uninfected drunken horse grass had better rates of seed germination under salt stress when PGPR was added. Endophyte infection alone improved plant germination under salt stress, so this effect may be compounded with PGPR [200]. This is likely due to superoxide dismutase, peroxidase, and catalase activity being increased. These enzymes are protectants of the plant cells from oxidative stress. Malondialdehyde, an indicator of oxidative damage, was also decreased in E+ soil. This suggests that the endophyte, along with PGPR, provide protections from the oxidative stress that high salt content could induce. The endophyte may also promote PGP bacteria under normal and drought conditions. A study carried out in 2016 found that E+ seedlings recovered faster from drought stress than E+ seeds that had been surface-sterilized and E- seeds. The E+ seeds had higher populations of PGP bacteria compared to E-, suggesting that the endophyte promotes this association [201]. Bacterial changes are likely due to changes in the root exudates that are modulated by the *Epichloë* endophyte. Changes in volatile organic compounds, flavonoids, and phenolic compounds have been shown under endophyte infection [119].

Many plants have mutualistic relationships with below-ground fungi called arbuscular mycorrhizal fungi (AMF). They help provide the plant with nutrients and can help with stressors such as drought and pathogens [202]. Some investigations into the effect of *Epichloë*-infected grasses on AMF find it greatly reduced. Interestingly, it has been found that neighbors of E+ grasses will have increased AMF infection. One mechanism by which this may occur is that the foliar endophytes require too many nutrients for the AMF to be sustained as well. The endophyte confers protections greater than that of the AMF in most cases, making it likely that the plant would choose to put its resources into it [203,204]. The change is likely due to endophyte-mediated changes to the rhizodeposits, which change the soil composition. There are also conflicting studies that show that *Epichloë* infection increased AMF, colonizing the roots. Vignale et al. [205] researched the AMF with *Epichloë* fungi in vitro by itself and in vivo with an association with *B. auleticus*. It was found that the endophyte exudates aided in AMF germination prior to root colonization, whereas the E- controls had no effect [205]. Stress conditions could also influence the grass–endophyte–AMF relationship. Zhong et al. [206] found that E+ drunken horse grass decreased AMF root colonization diversity under normal moisture conditions but increased AMF diversity under drought conditions. The mechanisms for this are unclear; however, it may be the grass allocating resources away from the endophyte when it is over-taxed. It appears that the 3-way association can work together to alleviate stress in the grass. Li et al. [207] found that E + perennial ryegrass with AMF colonization survived better under drought conditions by having increased P uptake, higher levels of photosynthesis, and osmoregulation. Similarly, Guo et al. [208] found that while both E+ perennial ryegrass and AMF colonized perennial ryegrass were more resistant to leaf spot, the combination of AMF and endophyte leads to the highest resistance. Low soil nutrients did not affect either fungi’s ability to mitigate the pathogen. Increased 3-glucanase activity and jasmonic acid activity were found in the resistant grass. Similarly, Bastias et al. [92] presented the idea that AMF can aid in plant defense through hormone signaling. This is called mycorrhiza-induced resistance and is thought to be mediated by JA. *Epichloë* enhancement of the JA pathway may help prompt this relationship with AMF that leads to increased pathogen resistance [92]. Whether AMF is being inhibited or enhanced, it is likely due to the root exudates produced by the grass–endophyte association. Similar to the research we have seen within the shoots of the plant, flavonoids and phenolic compounds are produced in higher amounts in the roots when *Epichloë* infections are present [206]. Phenolic compounds are generally thought to be anti-fungal and anti-bacterial; however, there are some studies showing increases in phenol production in association with AMF [209]. In addition to AMF, other fungi are affected by endophyte status. It was found that endophyte-infected tall fescue microbiomes had, overall, a very diverse fungal population at the genus level. There also was a distinct shift from basidiomycetes to ascomycetes fungi from E- to E+ grass [196]. Overall, the diversity of microbes will help contribute to the resilience of the plant under stress conditions. The exact host–endophyte association may lead to specific differences in the flavonoids and phenols being produced that would affect below-ground microbes in different ways, which may lead to the conflicting reports.

## 7. Discussion and Conclusions

In this review, we highlighted some of the major mechanisms by which cool-season grasses, *Epichloë* endophyte, and the rhizosphere work in harmony to mitigate abiotic and biotic stressors. This can be quite complex to untangle, but there are a few underlying mechanistic takeaways. *Epichloë* endophytes function to protect their grass hosts under stress through a combination of toxic alkaloid production, increased production and changes to enzymatic and non-enzymatic antioxidants, precursors to JA production, increased production and changes to VOCs, increased production of osmoregulatory substances, and increased production of other substances with vital physiological effects such as amino acids and carbohydrates. The upregulation and changes in the production of these substances are most likely due to the endophyte causing gene expression changes in the plant. Research has shown that endophyte infection causes many genes to be up and downregulated [35,36]. *Epichloë* fungal endophytes and cool-season grasses have a long evolutionary history together, in which they have been able to develop these molecular communications. Their co-evolvement has resulted in a change from pathogen to mutualist. One of the mechanisms responsible for this change is the production of antioxidants. An antioxidant is a substance that inhibits oxidation. These are often enzymes that mitigate the effects of ROS. ROS are natural byproducts of basic physiological functions like photosynthesis and respiration. When ROS accumulates, it causes damage at a cellular level. Antioxidants operate as part of the plant’s immune system. They function to help identify and respond to stress [52]. The *Epichloë* endophyte increases antioxidant production within grasses, which first operate to aid in mutualism and then aid in defense response. We see antioxidants playing an important role in defense against most abiotic and biotic stressors. Endophyte association also increases phenolic compounds in the plant, which are also associated with being an antioxidant and a plant defense. Phenols are found in higher amounts not only in the shoots of E+ grass, but also in the roots. In this way, they can have a direct effect on the soil microbiome. Another class of non-enzymatic antioxidants is flavonoids. These have also been shown to be produced in higher amounts in the shoots and roots of E+ grasses and can therefore also alter the microbial population [119]. Endophyte infection upregulates the production of other substances that have physiological implications, such as amino acids and sugars [68,72]. We see this particularly under drought and salt stress, where some of these substances have osmoregulatory properties. Amino acids also help with nutrient availability by changing the plant metabolism [68]. How the endophyte affects SA and JA pathways is somewhat unclear yet, but research has found that certain precursors to JA are produced in E+ grasses [91,92]. The increased production and changes to the production of VOCs can lead to host defense, communication, and phytohormone production [98]. In addition, differing VOCs are exuded by the roots of E+ grasses that would affect the microbiome [119]. Alkaloids can serve as antioxidants, but their main protection in grasses is to be toxic to herbivores, including insects and mammals. The combination of all these factors leads to a much higher performing plant under stress conditions (Figure 3). 

The utilization of *Epichloë* endophytes to enhance stress tolerance is an area that has been of interest. A few of the endophytes have already been manipulated to benefit their uses. Perennial ryegrass and tall fescue have had “novel” endophytes introduced that have a toxic alkaloid removed for enhanced livestock grazing performance [210,211]. The natural associations between endophytes and their grass hosts have largely been exploited by grass growers for different purposes throughout the world. However, utilizing these endophytic fungi beyond their host species poses more of a challenge. The host-specific nature of *Epichloë* makes it difficult to transfer the stress tolerance they confer to another non-natural host species. The key to unlocking this incompatibility likely lies in the plant-fungus feedback loop that allows for mutualism. A disruption in the pathway that creates pathogen response in the plant could lead to a compatible relationship. This would be most feasible within grass species. Work to identify fungi that maximizes host defenses and increases yield could achieve important results given many of the world’s most important crops, including corn, wheat, rice and barley, are members of the *Poaceae* family.

The changes that the endophyte induces in the soil microbiome are mixed. Studies have found conflicting information as to whether certain microbes are increased or decreased. The main takeaway from this is that every microbiome and E+ grass interaction is specific. Endophyte and host associations are very distinct in terms of compatibility. The different interactions between each fungus and host likely changes the ways in which root exudates are altered. In addition, microbiomes change drastically based on the environment. Taken together, it is impossible to make a blanket statement as to how *Epichloë* endophytes affect all microbes, except to say that there is some effect (Figure 3). Another takeaway is that manipulation of the microbiome through PGP bacteria and AMF fungi will probably only strengthen the fitness of the E+ grass and certainly will not be detrimental. 

These findings show that improved plant fitness goes far beyond just alkaloid production in E+ grasses. It also shows that the grass, endophyte, and microbiome is a very complex system that cannot be easily generalized (Figure 3). Nevertheless, these findings have implications for natural and agricultural systems that utilize the grass. It gives us an understanding of what goes into increasing grass performance under stress. It also gives us actionable items as far as biofertilization with PGP bacteria is concerned. More research into the molecular communication of this complex relationship to fully elucidate the mechanisms by which these organisms operate would provide even better insight that could potentially be utilized more broadly for agricultural systems.

## Figures and Tables

**Figure 1 microorganisms-09-02186-f001:**
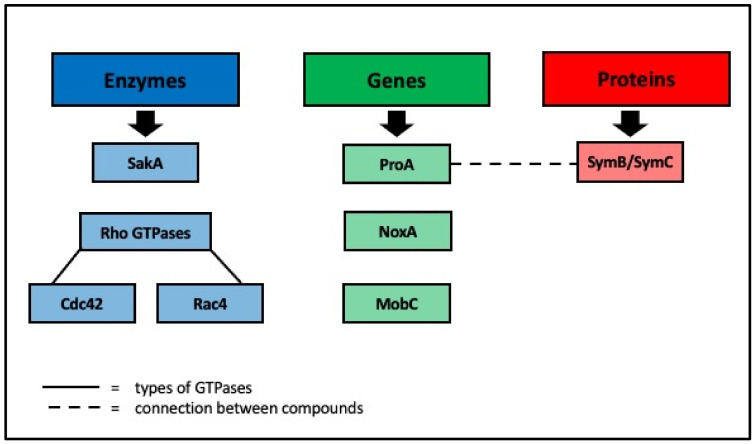
A visual representation of putative enzymes, genes and proteins necessary for *Epichloë* endophyte and grass mutualism. SakA and NoxA regulate ROS production. Cdc42 and Rac4 regulate intercalary and hyphal network growth. ProA works with SymB/SymC to regulate endophyte growth. MobC is necessary for the symbiosis of the endophyte and grass.

**Figure 2 microorganisms-09-02186-f002:**
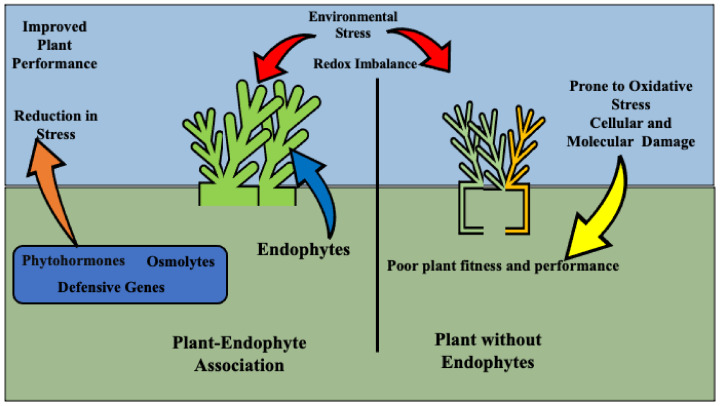
Environmental stressors impact on endophyte-infected and non-infected plants. Stress causes redox imbalances that lead to oxidative damage. In endophyte-free plants, there is little to mitigate this damage. Endophytes upregulate defensive genes and produce osmolytes and phytohormones that are protective against many stressors and reduce the damage done by redox imbalances. This improves the overall plant performance. Arrows represent effects on the plant and associations.

**Figure 3 microorganisms-09-02186-f003:**
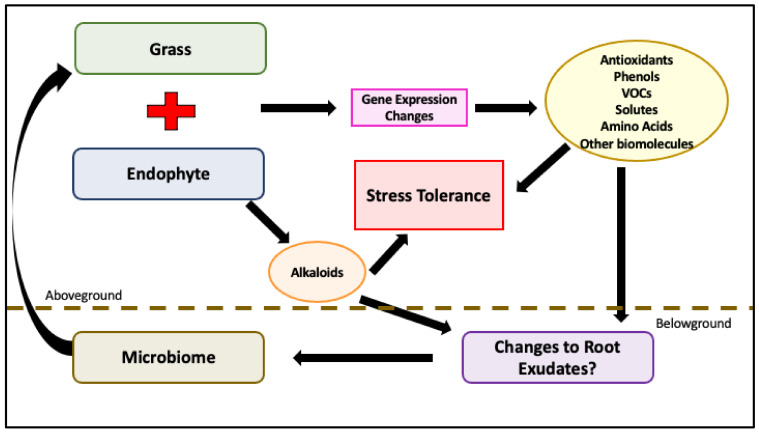
Diagram of proposed interactions between *Epichloë* endophytes, grass, stress and the microbiome. Grass infection with the endophyte leads to gene expression changes, which alters the production of a host of different compounds. The endophyte produces alkaloids. Alkaloids and the protective compounds produced lead to better stress tolerance of the grass. The same compounds and alkaloids may also have an effect on root exudates secreted by the grass into the soil. This could impact the microorganism present there, which in turn has a feedback effect on the grass. Arrows indicate how one change leads to another in the feedback loop.

**Table 1 microorganisms-09-02186-t001:** List of biotic and abiotic stressors, the potential tolerance/resistance that *Epichloë* infection could provide, the mediating biomolecules involved in resistance/tolerance and the mechanisms in which *Epichloë* infection works to combat the stressor.

Stressor	Tolerance/Resistance Potential	Mediating Biomolecules	Mechanisms	References
Drought	High	proline, glutamic acid, mannitol, loline alkaloids, oligosaccharides, antioxidant enzymes, ABA	- osmotic regulation via solutes and metabolites - increased root biomass - stomatal closure - decrease in sensitivity to drought through downregulation of drought genes - mitigation of oxidative damage	[44,45,46,47,48,49,50,51,52]
Salinity	High	K^+^, P, N, glycine betaine, enzymatic and non-enzymatic antioxidants	- Ion concentration regulation - Increase in net photosynthesis - osmotic regulation via amino acids and betaine	[53,54,55,56,57]
Heavy Metals	Medium	Enzymatic antioxidants, H_2_O, chlorophyll	- mitigation of oxidative damage - increased growth rate	[60,61,62,63]
Low Nutrients	High	G6DPH, NADPH, N-related enzymes, glutamine, amino acids, antioxidants, organic acids	- enzymatic regulation - amino acid metabolism regulation - organic acid metabolism regulation - mitigation of oxidative damage	[65,66,67,68,69]
Cold	Low	Fatty acids, proteins	- upregulation of fatty acid synthesis - protein turnover changes	[70,71,72,73]
Flood	Low	Antioxidants	- osmotic regulation - increased photosynthesis - mitigation of oxidative damage	[55,74,75,76,78]
Pathogens	High	alkaloids, antioxidants, proteins, salicylic acid, jasmonic acid, phenolic compounds, volatile organic compounds	- occupation of the ecological niche - antifungal protein production - mitigation of oxidative damage - increase salicylic acid production - increase jasmonic acid precursors - increase in precursors for shikimate and quinate pathways - phenolic compound and volatile organic compounds - deterrence of vectors	[51,79,80,81,82,83,84,85,86,87,88,89,90,91,92,93,94,95,96,97,98,99,100]
Nematodes	Medium	alkaloids	- potential change of root exudation - potential cell wall thickening	[101,102,103,104,105,106,107,108]
Insects	High	Peramine and loline alkaloids, jasmonic acid, volatiles	- alkaloid toxicity - increase in jasmonic acid precursors - potential multi-trophic interactions through volatile compounds - volatile olfaction for avoidance	[27,38,92,109,110,111,112,113,114,115,116,117,118,119,120]
Weeds	Medium	Phenolic compounds, syringic acid, myristic acid	- allelopathy via disruption of enzymes important for plant physiological processes	[121,122,123,124,125,126,127,128,129]
Animal Herbivory	High	Ergot alkaloids, indole-diterpene alkaloids	- neurotoxicity - metabolomic disruptions - gut microbiome disruptions	[130,131,132,133,134,135,136,137,138,139,140,141]

**Table 2 microorganisms-09-02186-t002:** *Epichloë* endophyte-infected grass influence on the accumulation of various compounds in the soil components.

Soil Component	E+ Effect	Reference
Organic C	Increased	[167,171]
Carbohydrates	Increased	[167]
Phenols	Increased	[168]
Organic Matter	Increased	[171]
Microbial Biomass C	Mixed Results	[165,172]
Mineralizable C	Decreased	[165,172]
Microbial Biomass N	Increased	[165]
Mineralizable N	Increased	[165]

## Data Availability

No new data were created or analyzed in this study. Data sharing is not applicable to this article.

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
