# Peer review of "Interaction between Grasses and Epichloë Endophytes and Its Significance to Biotic and Abiotic Stress Tolerance and the Rhizosphere"

_microorganisms, 2021, doi:10.3390/microorganisms9112186_

Round 1
Reviewer 1 Report
This is a well-written, very informative review. It is well structured and highly readable.
Author Response
Thank you for your comments!
Reviewer 2 Report
This is a comprehensive and well-structured review of the mutualistic association betwenn forage grasses and members of the Epichloe genus, which act as endophyte microorganisms. The significant of the study relies in the use of these grasses as feed for cattle. Thus, is important the understanding of those aspects that contribute to increase their growth and well-being. Despite the large volumen of information provided, the paper is easy to follow and inteconnect influence on different aspects and from different perspectives.
Just a few minor suggestions:
- Some scientific names are not in italics. On the contrary, Cocci and Bacilli (Page15) should not be in italics.
- The term “etc” and suspensive points are redundat.
- Bibliography is heterogeneus: jounal names sometimes are abbreviated and sometimes are complete; book chapters are not indicated as such, etc. Please, unify.
Author Response
Thank you for your comments. Please find all issues to be resolved.
Reviewer 3 Report
Dear Authors,
I reviewed your submission and unfortunately I think your paper is not suitable for publication in Microorganisms in its current form. Addressing the following issues can contribute to improve your manuscript.
Major issues (in decreasing order of significance)
Novelty. A Scopus search on "Epichloë" reviews on 09/30/2021 resulted in 26 review type documents in the last five years. These published reviews cover a large proportion of the aspects of your submission. Examples include but are not limited to drought (10.3389/fpls.2021.644731), heavy metals (10.3390/plants10030429), soil microbiology (10.1016/j.funeco.2021.101091), pathogenesis (10.1094/pdis-05-18-0762-fe), more detailed reviews on natural product chemistry and impact (10.3390/toxins11050302, 10.1080/00288233.2020.1785514), mechanisms beyond alkaloid production (10.1016/j.tplants.2017.08.005), and general overviews of the topic are also available (10.1021/acs.jafc.0c01396, 10.1016/j.fbr.2020.06.001, 10.1016/j.pbi.2018.01.010). What is more, the possibly cited recent original research is quite limited: the study cites only 19 studies from 2019-2021. Overall, I think this submission does not contain enough novelty to warrant publication in its current form. I suggest providing a detailed checklist on how this review is different from the ones mentioned above. Aspects that were not recently reviewed must be the focus of the revised paper.
There is no information on how the literature search was conducted. The search for "Epichloë" term results in 900+ articles in Scopus (abstract/title). The best would be the inclusion of search phrases and search engines / platforms used, as a supplementary material.
Ref. 179 is a preprint which did not pass any peer review, yet it is cited as if it was an article. Such references must be removed from the paper as long as they have a preprint status.
In many instances (e.g. L176), it is unclear from the manuscript's main text whether authors of the cited original research article did population-level studies, or plant-level studies. In the former case, the decrease of endophyte colonization would mean a population dynamic, in the latter, an eradication. This has to be explicitly stated in my opinion, as it is a very different scenario.
Genus name should be "Epichloë".
Minor issues (unsorted)
In my opinion, figures are of low standard. For example, the plant on the Figure 2 does not represent a "grass". I do not understand why you chose to use weird irregular shapes there either. Colors do not add any information (all figures).
- Charge should be superscript for ions (K+, Na+, and so on).
- L125: resveratrol is also a phenolic
- L57: secondary metabolites belong to the alkaloids
- L322: most phenolics are shikimates, but not all, there are also phenolic compounds biosynthesized via the polyketide pathway in plants (let alone fungi)
- L342: 11189 nematodes !?
- L326: claiming VOC inhibition in this form is too general
- L473: alkaloids are produced inside the plant
Best regards.
Round 2
Reviewer 3 Report
Dear Authors,
Thank you for your answers, I'm now rather convinced regarding the novelty issue. Please fix the following points, though.
1., L319: glycine and betaine (?)
2., Add taxonomic names to English names (Argentine Stem Weevil, drunken horse grass, etc.).
3., Despite the claim that it was fixed, ref. 201. still points to a preprint. State this in the main text.
Best regards.
Author Response
Dear Reviewer,
Thank you for your comments. Indeed the preprint was still there, I believe I was having an issue with EndNote, but it should be fixed now. I have ensure that all common names have a taxonomic name associated with it. I have double checked the term glycinebetaine from the original research article.
Thank you!
This manuscript is a resubmission of an earlier submission. The following is a list of the peer review reports and author responses from that submission.